# Impact of Shape Transformation of Programmable 3D Structures on UV Print Quality

**DOI:** 10.3390/polym16192685

**Published:** 2024-09-24

**Authors:** Matej Pivar, Deja Muck

**Affiliations:** Faculty of Natural Sciences and Engineering, University of Ljubljana, Snežniška 5, 1000 Ljubljana, Slovenia; matej.pivar@ntf.uni-lj.si

**Keywords:** 4D printing, thermal activation, UV inkjet printing, surface roughness, colour reproduction, shape transformation

## Abstract

The field of 3D and 4D printing is advancing rapidly, offering new ways to control the transformation of programmable 3D structures in response to external stimuli. This study examines the impact of 3D printing parameters, namely the UV ink thickness (applied using a UV inkjet printer on pre-3D-printed programmable structures) and thermal activation, on the dimensional and surface changes to high-stress (HS) and low-stress (LS) programmable samples and on print quality. The results indicate that HS samples shrink in the longitudinal direction, while expanding in terms of their height and width, whereas LS samples exhibit minimal dimensional changes due to lower programmed stress. The dynamic mechanical analysis shows that UV ink, particularly cyan and CMYK overprints, reduces the shrinkage in HS samples by acting as a resistive layer. Thicker ink films further reduce the dimensional changes in HS samples. Thermal activation increases the surface roughness of HS structures, leading to the wrinkling of UV ink films, while LS structures are less affected. The surface gloss decreases significantly in HS structures after UV ink application; however, thermal activation has little impact on LS structures. UV ink adhesion remains strong across both HS and LS samples, suggesting that UV inks are ideal for printing on programmable 3D structures, where the colour print quality and precise control of the shape transformation are crucial.

## 1. Introduction

Representing the next step in the evolution of manufacturing technologies, 4D printing combines the precision of 3D printing with the ability to program materials to change their properties over time in response to external stimuli. While 3D printing focuses on creating three-dimensional structures with high accuracy and repeatability, 4D printing enables the production of structures that can change their shape or properties over time under certain conditions. This opens up new possibilities in various disciplines, such as engineering, medicine, architecture, and others. Research in the field of 4D printing has rapidly advanced in recent years, particularly in the direction of using materials that can respond to changes in temperature, humidity, or other environmental conditions [1,2,3,4,5,6].

In the world of programmable structures, 4D printing offers limitless possibilities. Research in this innovative field often focuses on creating flat, programmable, multi-material structures that gradually change their shape under the influence of thermal activation [1,2,7,8]. This concept allows for the creation of structures that are both functionally and aesthetically appealing. The technology most commonly used for such printing is 3D printing, specifically the material extrusion technology known as FDM (fused deposition modelling). This technology uses various thermoplastics in filament form, making it one of the most widely used and cost-effective methods on the market [3,9,10,11,12,13]. Moreover, 3D printers operating with this technology are among the most accessible and affordable [1,2,14].

While many applications of 4D printing focus on creating origami structures [6,15,16,17] or simple everyday objects, such as phone stands and cases [2,8], an intriguing possibility lies in adding “new dimensions” through overprinting. This approach allows the enrichment of flat programmable structures with informative or decorative elements, such as rich graphics or additional information. This enables the creation of interactive structures that can include features such as QR codes and augmented reality [18,19,20]. Various printing technologies can be used for this purpose. For a larger series of identical products, conventional technologies such as screen printing and pad printing are more suitable, while digital technologies are better suited for a smaller series of products [21].

A significant advantage of overprinting 4D printed structures is the ability to personalize products for individual users. This means that products can be created that are fully tailored to individual preferences and needs. The most suitable technology for achieving this level of personalization is UV inkjet printing [22,23], which enables high-quality colour printing. Research has shown that overprinting 3D-printed structures with UV inkjet printing can increase the surface smoothness, which is important, for example, in printed electronics [24]. It also allows printing on surfaces with low relief, which is increasingly common in adapting artworks for blind and visually impaired people [25]. Of course, the print quality and the stability of the overprinted image are crucial in such products, with adhesion being of primary importance [26]. A study [22] found that adhesion in regard to UV inkjet printing on PLA thermoplastic surfaces was excellent without any surface pretreatment. It was also found that capillary phenomena occur during the printing process, due to the layered deposition of the filament, and that UV-LED curing is effective.

In addition, 3D printers that allow the printing of coloured products already exist on the market. Various 3D printing technologies are used for this purpose [27]. However, coloured products can also be printed using FDM technology in conjunction with UV inkjet printing, allowing graphics to be printed on the surface of each deposited filament layer. An example of such a printer is the XYZ printing da Vinci Color Mini 3D Printer [28]; however, these printers are more expensive than traditional FDM printers. When combining inkjet printing with material extrusion technology, it is also necessary to consider the colour management process and the inclusion of appropriate colour profiles to achieve accurate colour reproduction [28].

The quality of colour reproduction in inkjet printing is heavily influenced by the quality of the 3D-printed surface. Two key physical properties of the substrate that play a crucial role in achieving optimal colour reproduction are the surface roughness and surface gloss [29,30]. These properties are directly affected by 3D printing parameters and may change during the thermal activation of programmable 3D structures. Pretreating the surface of 3D-printed structures can help reduce roughness, increase smoothness, and even modify the surface gloss, all of which positively impact the final print quality and improve ink adhesion. Various methods are available for surface treatment, including laser treatment [31], hot air treatment [32], chemical treatment [33,34,35], and mechanical methods [36].

In addition to the surface roughness and gloss, ink adhesion to the substrate is a critical factor that greatly affects the quality of print reproduction. Without strong ink adhesion, high-quality reproduction cannot be achieved [37].

The importance of ink adhesion is also evident in other demanding fields. For example, article [26] investigates the use of UV inkjet printing technology in aerospace applications. It examines how different coatings on aluminium surfaces, using both solvent-based and UV-curing inks, affect adhesion, UV resistance, and print quality. The study shows that UV inkjet ink, when combined with a primer and a protective clear coat, delivers superior results in terms of adhesion, image clarity, and durability. This demonstrates its potential as a viable alternative to traditional aircraft painting methods, emphasizing the need for strong ink adhesion in environments where surface changes and external conditions can impact ink performance. Ink adhesion becomes even more critical in dynamic environments like 4D printing, where programmable 3D structures shrink and change shape during thermal activation, which can cause the ink to separate from the material’s surface.

In 4D printing, managing the interaction between ink adhesion and material deformation during shape transformations is particularly challenging. Previous research [2] has highlighted the importance of controlling residual stress during 4D printing to improve shape transformation accuracy. However, the effect of these residual stresses on the quality of colour overprinting, especially after thermal activation, has not been thoroughly explored. This gap underscores the need for further research into the interaction between ink adhesion, residual stress, and the stability of overprinted graphics during the transformation process. Understanding these dynamics is crucial to ensure that the ink remains adhered and that the image quality is maintained, even as the material undergoes structural changes.

This study focuses on the impact of thermal activation on multi-material, programmable 3D structures that were subsequently printed using a UV inkjet printer. The process of producing programmable 3D structures involved three stages. In the first stage, 3D printing was used to introduce stress into the thermoplastic materials, which is referred to as “programming while printing” [38]. In the second stage, high-resolution colour reproductions were printed onto the flat geometry of the programmable 3D structures using a UV inkjet printer. In the final stage, the programmable 3D structures were thermally activated to release the programmed stress and trigger their transformation into a new geometric form.

The programmable 3D structures consisted of active and passive parts. The active parts are the carriers of the programmed stress and serve to transform the programmable 3D structures, while the passive parts remain undeformed during thermal activation. We have investigated how different 3D printing parameters and thermal activation affect the quality of colour reproduction and the surface morphology of the material. We have found that thermal activation has a significant impact on the shape transformation of programmable structures, which in turn affects the print quality, including the roughness and gloss of the material.

We specifically investigated the effects of the programmed stress in the material on the stability of colour prints after thermal activation. We found that higher stress in the material leads to greater shrinkage of the filament and, consequently, to stronger deformation of the graphic elements. Additionally, we investigated the effect of the thickness of the UV ink film on the deformation and found that a greater thickness of the UV ink film leads to a higher level of resistance to material shrinkage, which has a negative effect on the transformation of the structures.

This study presents new possibilities for refining material stress programming and ink printing techniques, expanding the scope of 4D printing applications. Colour-printed, 3D programmable structures can be utilized in consumer products, such as customizable phone holders and hangers, as well as in interior design, didactic tools, and more. These applications combine functionality with aesthetics, offering innovative, adaptive solutions.

## 2. Materials and Methods

### 2.1. Materials and Printing Procedures

The stresses in the thermoplastic materials were programmed using a ZMorph VX 3D printer (ZMorph S.A., Poland). To produce the active part, different amounts of active layers of the PLA material were used, in combination with a passive layer of PRO-PLA material. However, to produce the passive part, different amounts of passive layers of PRO-PLA material were used, in combination with two active layers of PLA material (one at the bottom and one at the top). As the PLA material was always on the outside of the programmable 3D structure, the printed colour reproduction was only examined on the PLA material. The layer height was 0.2 mm in all cases. The 3D samples were produced specifically for the study, with low and high programmed stress. The 3D samples with low programmed stress (abbreviation; LS) were used to determine the effects of the activation medium and activation temperature on the properties of the UV ink, as this ensured that there was no shrinkage of the filaments and no deformation of the printed colour reproduction during thermal activation. The 3D samples with a higher level of programmed stress (abbreviation; HS) were used to determine the effects of the shrinkage of the deposited filaments on the quality of the colour reproduction. The 3D printing parameters for the production of the 3D samples are shown in Table 1.

The test images (Figure 1) for the colour reproduction quality analysis were printed on the 3D samples and the flat geometry of the programmable 3D structures using an Apex UV6090 UV LED flatbed printer (Apex, Shanghai, China). The printing was conducted with Nazdar 260 UV-LED inks, which were developed in order to achieve good adhesion and flexibility on a variety of commercially available printing materials, as well as excellent resistance to edge chipping and colour fading. The test images were printed in one printing pass, with linearisation only at a resolution of 1440 × 720 px. The four test images were carefully designed to evaluate all the important attributes of colour reproduction quality, using validated scientific methods. The first test image (Figure 1a) was designed to evaluate the geometric print quality of the graphic elements and the surface roughness. The second test image (Figure 1b) was designed to determine the CIELAB colour differences (ΔE*_ab_). The third and fourth test images (Figure 1c,d) were designed to determine the surface gloss, the adhesion of the UV ink, and its effect on the shape transformation of the programmable 3D structures.

Thermal activation of the 3D samples and programmable 3D structures was carried out in water at 80 °C for 15 min. After thermal activation, the samples were dried at 25 °C for 24 h to remove the moisture, in order to be ready for further analysis.

### 2.2. Material Characterization

#### 2.2.1. The Effect of the Printing Parameters on the Shape Transformation

The influence of the selected 3D printing parameters on the programmable state of the 3D-printed HS and LS samples was determined by measuring the dimensional changes to the 3D-printed samples after thermal activation. The dimensions were measured before and after thermal activation using a digital calliper (Mitutoyo, Sakado, Japan). The dimensions of the designed 3D samples were 4.5 (X) × 100 (Y) × 3 (Z) mm. The percentage of deformation to the sample dimensions (ε) was calculated according to Equation (1).
ε = (L_f_ − L_i_)/L_i_ × 100%(1)
where L_i_ is the initial dimension before thermal activation and L_f_ is the final dimension of the 3D-printed sample after thermal activation [39,40]. Five 3D-printed samples were tested for each group of samples.

The influence of UV ink on the programmable state of the 3D-printed samples was determined using dynamic mechanical analysis (DMA), with a DMA Q800 instrument (TA Instruments, New Castle, DE, USA). The influence of two UV ink films of different thicknesses on the programmed state in the longitudinal direction was determined by measuring the change to the initial dimensions of the 3D samples during heating. The analysis was performed only for the HS samples in tensile mode and at a constant low load of 0.01 N [2]. The HS samples were selected because they achieve greater dimensional changes during thermal activation. The 3D samples were printed once, on the top and bottom, with cyan ink, with a tonal value (TV) of 100%, and a second time with an overprint in terms of all the process colours (CMYK), resulting in a black colour, with a total ink coverage (TIC) of 400%. The results were compared with a reference sample, showing the dimensional change to the 3D sample without printed UV ink.

The influence of UV ink on the shape transformation of programmable 3D structures was determined experimentally, as a function of the number of active layers. The number of active layers was 6, 8, and 10, and they were printed with the same printing parameters as the HS samples to create a high-stress structure. The programmable 3D structures were printed with cyan and CMYK overprints, on the top and bottom surfaces, in the same way as for the DMA analysis, and were thermally activated. To determine the radius of curvature, photographs of the programmed 3D structures were taken after thermal activation, with a Nikon D750 (FX) camera (Nikon Europe, Amsterdam, The Netherlands), and the outer radius of curvature was determined using the Digimizer software 6.3.0 (MedCalc Software Ltd., Ostend, Belgium) [2]. The measurements were compared with a reference radius of curvature of programmable 3D structures without printed UV ink. For each group of samples, 10 programmable 3D structures were tested.

#### 2.2.2. Surface Properties

Two important and related physical properties of the substrate surface that must be considered for optimum colour reproduction quality are the surface roughness and surface gloss [29,30]. Both surface properties are influenced by the 3D printing parameters and can change during the thermal activation of programmable 3D structures.

The roughness of the unprinted and printed surfaces of the 3D-printed samples with UV ink was measured before and after thermal activation, using a TIME 3200/3202 (TR200) contact roughness profilometer, according to ISO 4287 [41]. For all the measurements, the cut-off distance was set to 2.5 mm, depending on the surface roughness of the substrate and the printed UV ink. The roughness profile (R-profile) and the average roughness values (Ra), which provide a good general description of the height variations in the surface [30], were analysed in perpendicular (measuring angle 90°) and parallel (measuring angle 0°) directions to the deposited filaments in the five 3D-printed samples.

The gloss of the unprinted and printed surfaces of the 3D-printed samples with UV ink was measured before and after thermal activation, using a Rhopoint IQ 20/60/85 glossmeter, in accordance with ISO 2813 [42]. The surface gloss was measured for all three measuring geometries (20, 60, and 85°) in the parallel direction to the deposited filaments. The measurements at a universal angle of 60° were used for the basic surface analysis of the 3D-printed materials. For a better resolution on surfaces with low gloss, the measurements at an angle of 85° were used for the comparative analysis of the printed UV ink and the surface of the material before and after thermal activation. The surface gloss was measured on five samples for each HS and LS group of 3D-printed samples.

#### 2.2.3. Ink Adhesion

The adhesion of the ink to the substrate is an important parameter that determines the quality of the print reproduction. Without good ink adhesion, high-quality reproduction cannot be achieved [37]. In 4D printing, good ink adhesion is even more important, as programmable 3D structures shrink and change shape during thermal activation, which can cause the ink to detach from the surface of the material. 

To determine UV ink adhesion before and after thermal activation on the HS and LS samples, a cross-cut test was carried out using the Byko-cut universal device (BYK-Gardner GmbH, Geretsried, Germany), in accordance with ISO 2409 [43]. The cuts were made at an angle of 45° to the direction of the deposited filaments, using a cutting tool, with a blade spacing of 1 mm, depending on the UV ink thickness. The cross-cut pattern was visually inspected using a Dino-Lite AM4113ZT digital microscope (AnMo Electronics Corporation, Taiwan, China) and compared with the ISO classification.

#### 2.2.4. Colour Differences

The CIELAB colour differences (ΔE*_ab_) were used to evaluate the variation in the appearance of the process colours and the colour degradation of the material after thermal activation. The CIELAB colour values (L*, a*, and b*) were measured before and after thermal activation on the HS and LS samples, using a Datacolor Spectro 1050 spectrophotometer (Datacolour, Lawrenceville, NJ, USA), under D50 illumination, with a 2° standard observer, with specular included, and with a d/8° measurement geometry. The CIELAB colour values were measured on 5 samples and the colour differences were calculated using CIEΔE*_76_ Equation (2).
ΔE*_ab_ = [(ΔL*)^2^ + (Δ a*)^2^ + (Δ b*)^2^]^1/2^(2)
where ΔE*_ab_ stands for the colour differences [/], ΔL* for the differences in lightness, Δa* for the red–green differences, and Δb* for the yellow–blue differences [26,44].

#### 2.2.5. Evaluation of the Geometric Print Quality of Graphic Elements 

Visual inspection of the microscopic images and image analysis were used to examine all the relevant image quality attributes that determine the geometric print quality of the printed graphic elements, such as the edge raggedness (edge noise), physical gain, dot roundness, and the uniformity of the ink density [45,46]. The graphic elements printed on the HS and LS samples were captured using a Dino-Lite AM4113ZT digital microscope (AnMo Electronics Corporation, Taiwan) at 50× magnification and a 1280 × 1024 px resolution, using a polarising filter to eliminate the surface gloss of the UV ink. The captured microscopy images were then processed and analysed with ImageJ 1.54 software (open source), using various validated scientific methods.

The edge raggedness (edge noise) and physical gain were determined for 0.3 mm wide lines printed at different angles, depending on the direction in which the filaments were deposited in the 3D-printed samples. The aim was to determine whether the different orientation of the fine line elements to the direction of the deposited filaments affected their quality. The edge raggedness was determined by evaluating the line perimeter [45] and the physical gain was determined by evaluating the line area. Five measurements were taken for each line orientation and the average values were compared with reference values (ideal) from a digital document.

The uniformity of the ink density was determined by visual assessment of a topographic projection created using the Interactive 3D Surface Plot (ImageJ) tool, which converts the ink density to a proportional height to see how thick the ink film is [45]. The topographic projection shows the uniformity of the ink density over the entire surface of the captured image, which relates to the thickness of the ink film and the properties of the substrate [47].

In order to determine the influence of the surface properties of the samples and the thermal activation on the deformation of the graphic elements, dot roundness analysis was carried out, which describes the change in the dot shape in relation to a geometrically perfect circle. The roundness value for the geometrically perfect circle (ideal dot) is defined by 1. Dots with a diameter of 0.8 mm were chosen for the analysis because they achieve a higher roundness value compared to smaller dots [30] and, therefore, more accurately describe the deformation of the shape due to the effect of thermal activation. The dot roundness was determined according to Equation (3) [30,48].
Roundness = [4 × area/(π × major_axis^2^)](3)

The roundness was determined as the average value of the roundness of a total of 45 dots before and after thermal activation for each group of HS and LS samples.

#### 2.2.6. Surface and Structure Morphology

A scanning electron microscope (SEM), 6060 LV (Jeol, Tokyo, Japan), was used to observe the surface and cross-sectional morphology of the 3D samples printed with UV ink. The surface morphology of the printed UV ink was observed on the HS samples before and after thermal activation to determine how thermal activation in hot water and filament shrinkage affect the material and printed UV ink. The morphology in terms of the structure and the thickness of the printed UV ink were observed through a cross-section. The thickness of the UV ink film on the HS samples printed with cyan ink, with a tonal value of 100% and CMYK overprints with a total ink coverage of 400%, was measured at 50 spots, using ImageJ software.

## 3. Results

### 3.1. Results of the Effect of Printing Parameters on the Shape Transformation

The average values and standard deviation of the dimensional changes to the 3D-printed samples after thermal activation are listed in Table 2. A maximum shrinkage of 14.63% was measured in the longitudinal direction of the deposited filaments in the HS samples. The shrinkage of the filaments in the longitudinal direction causes the 3D-printed samples to expand by 13.10% in height and 5.19% in width. The LS samples show similar behaviour, but the dimensional changes are on average less than 0.3% in the longitudinal and transverse directions and less than 0.6% in the height direction. The smaller dimensional changes to the LS samples are influenced by several factors. One factor is the isotropic structure of the 3D-printed samples, which is created by a rectilinear infill pattern. Wang et al. [49] reported that a rectilinear infill pattern eliminates the differences in the shrinkage rate and, thus, minimises the transformation. Other factors that reduce the dimensional changes to 3D-printed samples are the 3D printing parameters. The higher the 3D printing temperature and build plate temperature and the lower the printing speed, the less prestress is stored in the material and the lower the dimensional changes to the 3D printed sample [39,50,51,52].

The DMA analysis (Figure 2a) shows that all the samples initially expand in the range of the glass transition temperature of the PLA material (approx. 61 °C) and then begin to shrink in the range from 67.52 to 70.15 °C. The extent of the shrinkage of the 3D samples is influenced by the printed UV ink. The UV ink printed on the top and bottom of the 3D samples resists and inhibits shrinkage. The cyan ink reduces shrinkage by 3.49% and the CMYK overprints by 11.31%, compared to the final shrinkage of the unprinted reference sample of 19.07%. The average cyan film thickness at a tonal value of 100% is four times lower than the CMYK overprint film thickness at a total ink coverage of 400%. The thickness of the printed UV ink film, as detailed in Section 3.7, was measured with ImageJ software, using the images taken with the SEM microscope. Both samples printed with UV ink start to elongate slightly when they reach maximum shrinkage and then stabilise. In contrast, the 3D reference samples stabilise earlier.

The analysis of the experimental determination of the radius of curvature (Figure 2b,c) of the programmable 3D reference structures showed that the radius of curvature increases linearly (R² 0.998) with the number of active layers, which means that the programmable 3D structures achieve a smaller level of transformation. The 3D samples printed with different thicknesses of UV ink film behave similarly, but achieve a smaller level of shape transformation than the reference sample due to the effect of the printed UV ink. The results can be compared with the DMA analysis, as the behaviour is similar. CMYK overprints inhibit and reduce the shape transformation of the programmable 3D structures on average more than printed cyan ink, regardless of the number of active layers. From this, it can be concluded that the thicker the UV ink film, the more it resists the shrinking of the programmed filaments and the bending of the programmed 3D structures during thermal activation, which has a negative effect on their transformation. This leads to new insights and opens up new avenues of research related to the finding that UV inks printed in a sufficiently thick film can be used as a passive layer to control the transformation of programmable 3D structures. When developing programmable 3D structures for printing involving colour reproduction with UV inkjet printers, the thickness of the UV ink film and the number of active layers must be considered.

### 3.2. Surface Roughness Measurements

The analysis of the surface roughness (Table 3 and Figure 3a,b) of the 3D-printed samples showed that the roughness in the perpendicular direction (90°) is significantly higher than in the parallel direction of the deposited filaments in all the cases examined. The large variation in surface roughness is due to the anisotropic deposition of the extruded filaments in the same layer. The R-profile measured in the perpendicular direction (Figure 3a) shows an even distribution of peaks and valleys. The partially smoothed peaks represent the peaks of the deposited filaments, while the sharp valleys represent the channels of the neighbouring deposited filaments, where they fuse together. The R-profile measured in the parallel direction (Figure 3b) represents the valley measured in the longitudinal direction between two neighbouring filaments, with no peaks and no major deviations in the surface roughness. 

The average roughness values and standard deviation show that the printed UV ink increases the roughness measured parallel to the deposited filaments for both groups of samples, while this cannot be deduced in the perpendicular direction due to the large standard deviation in terms of the measurements. In the parallel direction of the deposited filaments, the roughness (Figure 4a,b) increases due to the large amount of ink, which is between 5 and 15 μm [30], and the uneven distribution of the large ink droplets that do not penetrate the substrate. In the perpendicular direction of the deposited filaments, the R-profile shows that the partially smoothed peaks generally become more irregular. The peaks of the filaments become less uniform and jagged, while the depth of the valleys between the filaments decreases (Figure 5b). The reduction in the depth of the valleys is due to the capillary action of the channels between the deposited filaments, which draw the liquid UV ink into the gap before it is cured by the UV light. Roach et al. [24] report that voids on the surface of 3D-printed parts cause poor geometry of the printed elements and cracks in the printed ink that affect the print quality.

Further investigations show that thermal activation has a major influence on the surface roughness of the printed UV ink on the HS sample, measured parallel to the deposited filaments. The average surface roughness values increase from 1.561 (±1.561) µm to 5.292 (±0.571) µm. The main reason for this change is the programmed stress in the deposited filaments, which triggers filament shrinkage in the longitudinal direction during thermal activation. In Section 3.1, it was determined that the dimensions of the 3D samples decrease by 14.63 (±0.396) % in this direction. The roughness measurements and the R-profiles (Figure 6a,b) show that the shrinkage of the filaments in the longitudinal direction causes the wrinkling of the printed UV ink. For the other samples, where the shrinkage of the filaments is not as strong, there are no major differences in the surface roughness due to thermal activation.

### 3.3. Surface Gloss 

The surface gloss at a universal measurement geometry of 60°, measured in the longitudinal direction of the deposited filaments, shows that the surface of the PLA material is defined as semi-glossy in both cases (22.56 GU for the HS and 10.14 GU for the LS samples). The surface gloss of the HS samples is on average 14.42 GU higher than that of the LS samples. The differences in surface gloss are caused by different 3D printing parameters that affect the surface of the 3D-printed samples. As reported by Žigon et al. [53], surface gloss is highly related to surface roughness. As the surface roughness increases, the gloss decreases, resulting in a dull or lustreless surface [54]. The HS samples in the longitudinal direction of the deposited filaments (Table 3) have a lower average roughness of 0.767 (±0.232) µm than the LS samples of 1.045 (±0.180) µm, resulting in a higher surface gloss. 

For further investigations, the 85° measuring geometry with a larger measuring spot was used, as it is more suitable and more accurate for determining the gloss of matt surfaces [54]. The comparison of the gloss measurement between the unprinted and printed surfaces of the samples with UV ink showed that the UV ink has an influence on the surface gloss (Figure 7). After printing with UV ink, the surface gloss of the HS samples is reduced by an average of 42.3 GU and that of the LS samples by 42.5 GU. In both cases, UV printing increases the average surface roughness (Table 3), which leads to a significantly lower surface gloss.

Thermal activation has no significant effect on the surface gloss of the material itself and on the UV ink printed on the LS samples. The gloss reduction is between 4.62 GU for the HS samples, 3.34 GU for the LS samples, and 2.02 GU for the UV ink printed on the LS samples. The measurements of the surface roughness (Table 3) showed that thermal activation leads to shrinkage of the material, causing the UV ink to wrinkle and the light to be reflected more diffusely. 

### 3.4. Printing Ink Adhesion

The analysis of UV ink adhesion revealed three important findings for 4D printing. The adhesion of the UV ink to PLA material is good without treating the surface of the material. The activation medium and the activation temperature have no influence on the ink adhesion, as there is no visible difference in the adhesion on the LS samples. Even a 14.63% shrinkage of the HL samples in the longitudinal direction has no influence on the UV ink adhesion. The optical images (Figure 8) of the cross-cuts show that the adhesion of the UV ink to the PLA material achieves ISO classification 0. The cut edges are completely smooth and none of the squares in the grid are detached. In all cases, the adhesion of the UV ink is the same on all the samples.

### 3.5. Results of Colour Differences

The colour differences (∆E*_ab_), shown in Figure 9, represent the difference between the printed colours and the colour of the unprinted material surface before and after thermal activation. In all cases, the average colour difference for the LS samples were between 1.4 and 1.6. This means that the activation medium and temperature do not have a major influence on the appearance of the colours. These colour differences are very small and are normally indistinguishable to an inexperienced observer, but they are visible to the trained eye when the samples are placed side by side [26,44]. A slightly higher deviation in the average values between the colours was observed in the HS samples. The values of the colour differences range from 0.6 to 2.6. The largest colour differences of 2.6 were observed for the colour cyan. These colour differences are obvious to the untrained human eye (2 < ∆E*_ab_ < 3.5) [44], which means that the shrinking of the HS samples changes the appearance of the colour. The other colour differences are small and cannot be distinguished by an untrained eye.

### 3.6. Results of Evaluation of the Geometric Print Quality of Graphic Elements

The visual evaluation of the microscopic images (Figure 10) showed that the UV ink is absorbed into the channels along the deposited filaments, as reported by Wögerer et al. [22], which was also confirmed by some of the roughness measurements. UV inkjet printing is characterised by the fact that the UV ink changes almost instantly from a liquid to a solid form when exposed to sufficient UV radiation [55]. In our case, the capillary action is stronger and faster, so the UV ink is absorbed into the channels before it cures, making the edges of the graphic elements slightly less pronounced and serrated. Excessive edge serration on fine elements can affect the clarity and visibility of the elements, as well as the fidelity and legibility of the text [45]. Physical gain can also occur, which has a negative effect on the appearance of the colour reproduction.

The direction of the capillary action (Figure 10b) depends on the direction of the deposited filaments in the active part, which controls the direction of the shape transformation of programmable 3D structures and has no function in regard to the passive parts. The analysis of the image quality of the 0.3 mm thick lines printed at 0, 45, and 90° to the direction of the deposited filaments has shown (Table 4) that orientation has an influence on the degradation of the edges, which manifests itself as smearing of the ink and affects legibility. In all cases, the perimeter values of the lines exceed the reference values. It can be assumed that higher line perimeter values indicate greater line raggedness [45]. The highest raggedness was observed when the lines were printed perpendicular (90°) to the direction of the loaded filaments, while the lowest raggedness was measured for the lines printed parallel (0°) to the loaded filaments, as no capillary action occurs in this direction. The surface area measurements showed that there were no significant differences in the surface area between the lines printed at different angles, but all the measurements exceeded the reference values. This indicates that a physical gain has occurred due to the spreading of the liquid UV ink before curing, due to the low surface tension of the material [56] and the scattering of a very small satellite ink droplet at the edge of the fine element [30].

The topographic projection (Figure 10c) shows that the capillary action also causes a macro print mottle, which is characterised by an uneven appearance in the solid area printed over the entire surface, is usually caused by uneven ink absorption into the substrate and plays an important role in print quality Ž46]. The uniformity of the printed image was analysed by visual assessment of the topographical projection of the solid area printed with black (100% TV) UV ink. During the topographic conversion, white spots on the microscopic image were smoothed out, as they were due to the reflection of the ink gloss when the image was taken with a digital microscope, and are not a relevant representation of the optical density of the UV ink. The topographical projection clearly shows a very unevenly printed entire surface. The nonuniformity in ink density is reflected as parallel black lines that have a higher optical density than the rest of the image due to ink absorption in the channels between the deposited filaments, which affects the appearance of the printed colour reproduction.

The dot roundness values obtained are shown in Table 5. The differences in the surface roughness between the samples, resulting from the different 3D printing parameters, have no significant influence on the roundness of the dots. The dots have practically the same roundness value without thermal activation. However, thermal activation appears to have a greater influence on the roundness of the dots in the HS samples. The roundness of the dots decreases from 0.954 to 0.910, while the roundness of the dots in the LS samples does not change. The visual evaluation of the microscopic image (Figure 11) shows that the dots shrink in height and expand slightly in width, which can be attributed to the thermal shrinkage behaviour of the filament. The same tendency can also be seen in the distance between two neighbouring dots. The distance between the dots in terms of the height (longitudinal direction of the filaments) decreases, while the distance between the dots in terms of the width (transverse direction of the filaments) increases. From these results, it can be concluded that thermal activation deforms the graphic elements of colour reproduction printed on the active parts of the programmable 3D structures depending on the degree of programmed stress and the direction of the deposited filaments, which must be taken into account in the pre-press stage of colour reproduction.

### 3.7. Visualization of Surface and Structural Morphology 

The observed SEM images of the cross-sections (Figure 12a,b) were used to determine the thickness of the printed UV ink film on the surface of the 3D-printed samples. Due to the non-uniform thickness of the UV ink film, 40 measurements were performed for each sample. The average film thickness of the cyan ink at a tonal value of 100% on the HS sample in one pass, which can be clearly seen in Figure 12a, is 13.64 (±5.266) µm, while the average film thickness of the CMYK overprint at a total ink coverage of 400% in one pass is 55.36 (±17.654) µm, which is four times thicker than that of the cyan ink. The measurements show that the ink is not absorbed into the substrate during UV printing, but remains cured on the substrate surface, which is characteristic of UV inkjet printing. The morphology of the cross-sections shows that the thickness of the CMYK prints in the channel between the deposited filaments is much greater than at the tip of the individual filaments. The spreading and overlapping of the droplets in the channel between the filaments, which affect edge ragging, the physical dot gain, and uneven ink density, can be clearly seen in Figure 13a,b. Capillary action draws each individual ink droplet into the channel, where it spreads and deforms in the direction of the channel and overlaps with other droplets, in relation to other droplets on the surface of the filament, which have a more regular geometric shape. The morphological analysis of the surface of the PLA material (Figure 13b) shows that thermal activation in hot water changes the surface of the material. Before thermal activation, the surface of the material is smooth and without irregularities, but after thermal activation, the material swells and pores appear on the surface. It is assumed that the changes in the surface morphology are a combination of the material shrinkage and the absorption of hot water into the PLA material during thermal activation. 

The comparison of Figure 14a,b shows the effects of the shrinkage of the programmable filaments on the surface of the printed UV ink. During thermal activation, the UV ink shrinks with the deposited filaments and becomes wrinkled. The wrinkling of the ink is consistent with the measurements of surface roughness and gloss on the HS samples. The wrinkling leads to increased roughness and reduced gloss. No wrinkling of the UV ink was observed on the LS samples. However, no cracking of the UV ink or delamination of the surface was observed either, indicating that UV inks are suitable for printing on programmed 3D structures.

## 4. Conclusions

This study has illuminated the significant impact that thermal activation has on the shape transformation and print quality of multi-material, programmable 3D structures. The results clearly demonstrate that the interplay between the stresses programmed into the material and the application of UV inkjet printing introduces complex dynamics that affect the final printed product.

The key findings indicate that thermal activation causes substantial dimensional changes in the high-stress (HS) samples, leading to filament shrinkage that not only affects the structure’s transformation, but also impacts the surface roughness and gloss. The study also revealed that the thickness of the UV ink film plays a critical role in resisting these dimensional changes, with thicker ink films resulting in less pronounced transformation of the programmable 3D structures.

Furthermore, the research underscores the importance of both the material properties and the printing parameters to ensure that the desired shape transformations do not compromise the print quality. The capillary action observed during the UV inkjet printing process, particularly in relation to the anisotropic deposition of the filaments, highlight the need for the careful control of ink deposition to avoid issues such as edge raggedness, print mottle, and non-uniform ink density.

Moving forward, future research should focus on refining the balance between material stress programming and ink application to enhance the stability and predictability of shape transformations in 4D-printed structures. Additionally, exploring new materials and advanced post-processing techniques could further mitigate the challenges posed by filament shrinkage and ink deformation, ultimately paving the way for more robust and versatile applications of 4D printing in various industries.

## Figures and Tables

**Figure 1 polymers-16-02685-f001:**
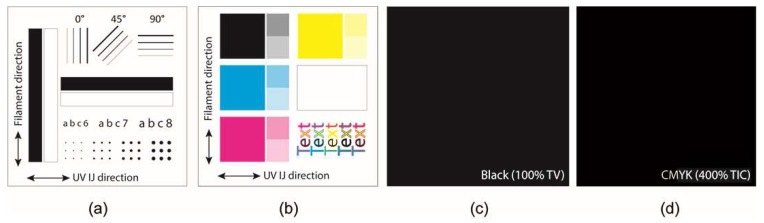
Test images for evaluating the print quality.

**Figure 2 polymers-16-02685-f002:**
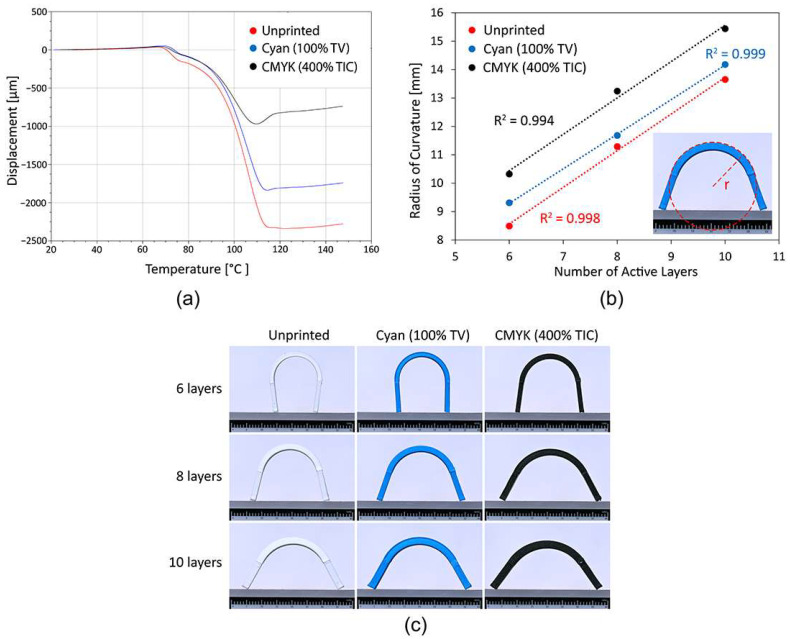
(**a**) DMA measurements of dimensional changes in 3D-printed samples. (**b**) Measured results of the radius of curvature as a function of the number of active layers and the different thicknesses of the UV ink films. (**c**) Visual comparison of the radius of curvature as a function of the number of active layers and the different thicknesses of the UV ink films.

**Figure 3 polymers-16-02685-f003:**
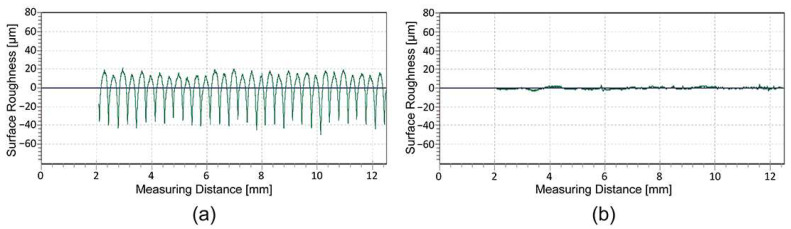
Comparison of the R-profiles: (**a**) measured perpendicular and (**b**) parallel to the deposited filaments.

**Figure 4 polymers-16-02685-f004:**
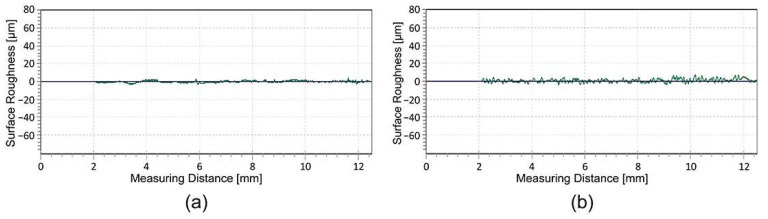
Comparison of the R-profiles measured parallel to the deposited filaments: (**a**) before and (**b**) after UV inkjet printing on the same HS sample.

**Figure 5 polymers-16-02685-f005:**
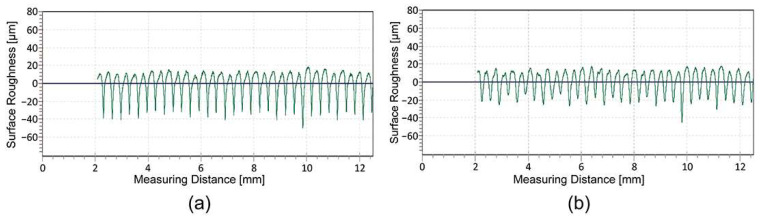
Comparison of the R-profiles measured perpendicular to the deposited filaments: (**a**) before and (**b**) after UV inkjet printing on the same HS sample.

**Figure 6 polymers-16-02685-f006:**
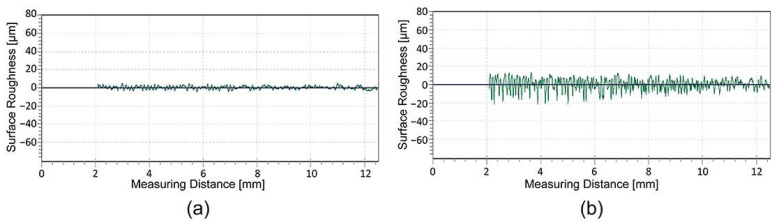
Comparison of the R-profiles measured parallel to the deposited filaments in the printed UV ink on the same HS sample: (**a**) before and (**b**) after thermal activation.

**Figure 7 polymers-16-02685-f007:**
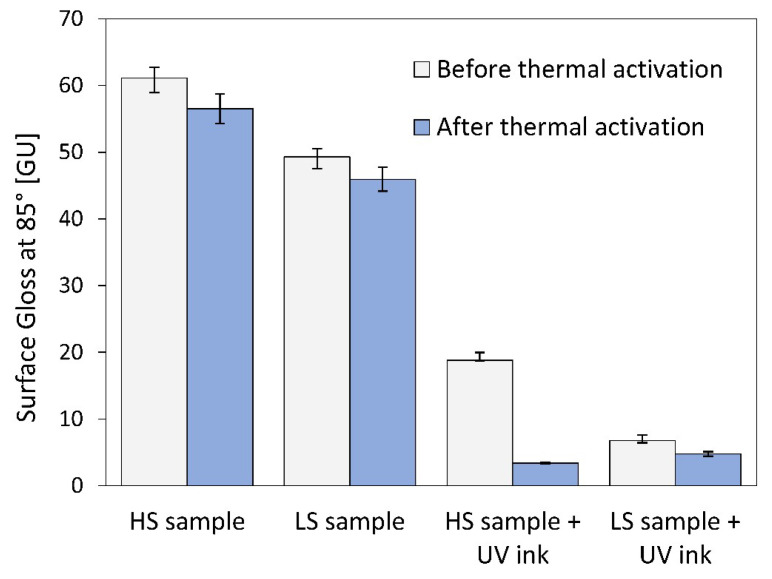
Surface gloss measurements, with a measuring geometry of 85°.

**Figure 8 polymers-16-02685-f008:**
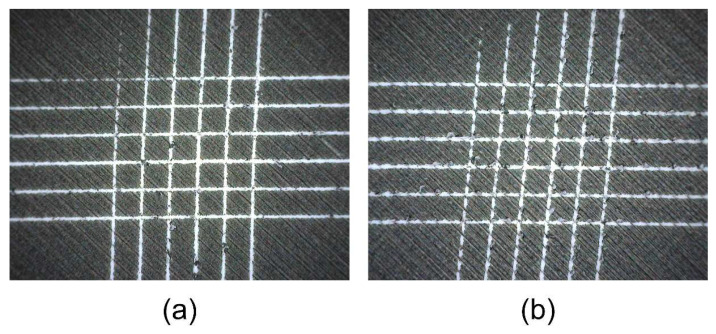
Images of the cross-cut pattern after thermal activation: (**a**) HS sample and (**b**) LS sample.

**Figure 9 polymers-16-02685-f009:**
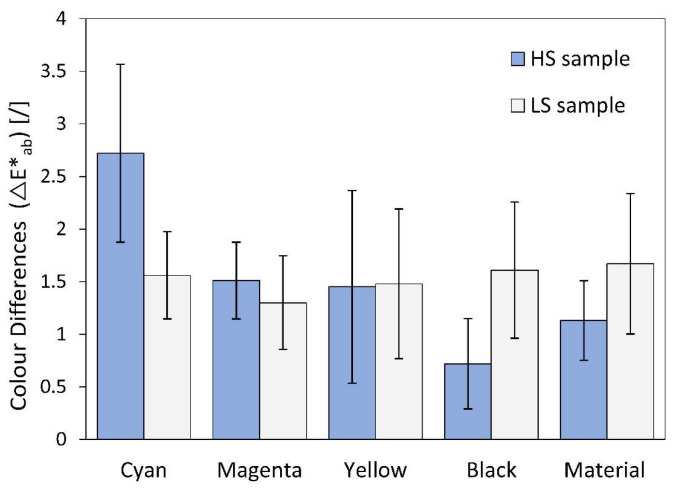
Colour difference values.

**Figure 10 polymers-16-02685-f010:**
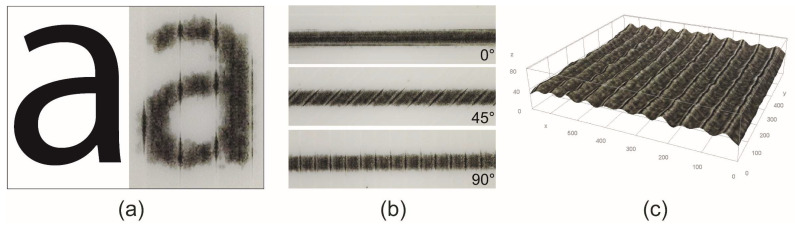
(**a**) Visual comparison of the printed “a” character (8 pt) with the reference. (**b**) Visual comparison of the 0.3 mm thick lines printed at different angles to the deposited filaments. (**c**) Topographic projection of the solid ink density.

**Figure 11 polymers-16-02685-f011:**
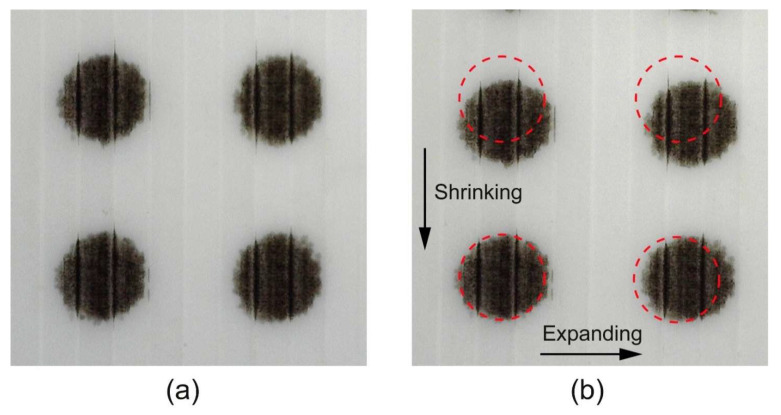
Visual comparison of the position and shape of the printed dots, with a diameter of 0.8 mm, on the same HS sample: (**a**) before and (**b**) after thermal activation.

**Figure 12 polymers-16-02685-f012:**
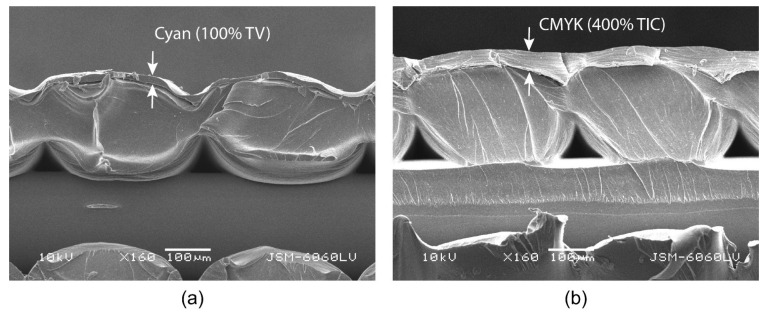
SEM images of a cross-section of different thicknesses of UV ink: (**a**) cyan at a tonal value of 100% and (**b**) CMYK overprints at a total ink coverage of 400%.

**Figure 13 polymers-16-02685-f013:**
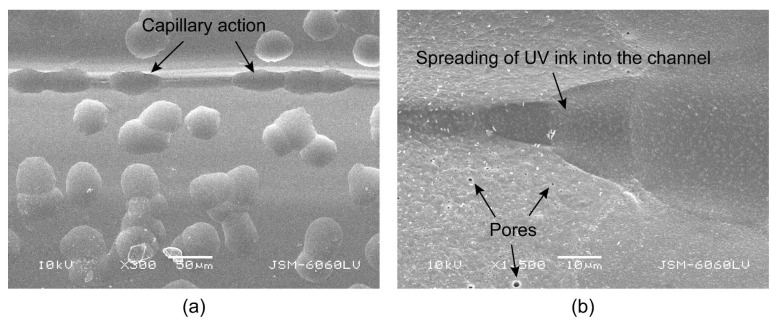
(**a**) Capillary action and uneven distribution of droplets of UV ink within the same halftone area. (**b**) Capillary action and morphology of material surface after thermal activation.

**Figure 14 polymers-16-02685-f014:**
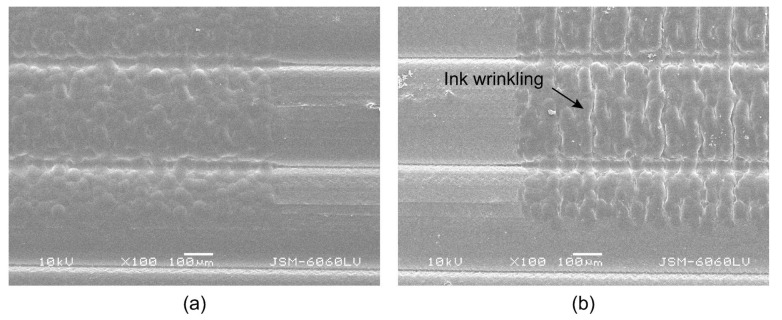
SEM images of the printed ink surface: (**a**) before and (**b**) after thermal activation.

**Table 1 polymers-16-02685-t001:** 3D printing parameters.

Sample	Printing Temperature [°C]	Build Plate Temperature [°C]	Printing Speed [mm/s]	Fan Speed [%]	Infill Pattern
HS	195	30	50	100	Aligned Rectilinear
LS	225	80	20	0	Rectilinear

**Table 2 polymers-16-02685-t002:** Average values and standard deviation of the dimensional strain on the 3D-printed samples.

Sample	Length [%]	Width [%]	Height [%]
HS	–14.63 (±0.396)	5.19 (±0.432)	13.10 (±0.174)
LS	–0.29 (±0.059)	0.16 (±0.302)	0.53 (±0.447)

**Table 3 polymers-16-02685-t003:** Average values and standard deviation of surface roughness.

Sample	Measuring Direction	Ra [µm]			
		Material before Activation	Material after Activation	UV Ink before Activation	UV Ink after Activation
HS	Parallel	0.767 (±0.232)	1.507 (±0.972)	1.561 (±0.232)	5.292 (±0.571)
	Perpendicular	10.647 (±0.980)	12.096 (±1.021)	11.386 (±1.196)	11.430 (±0.307)
LS	Parallel	1.045 (±0.180)	1.110 (±0.244)	2.175 (±0.289)	2.905 (±0.367)
	Perpendicular	12.477 (±4.164)	12.079 (±1.869)	13.582 (±3.608)	13.634 (±2.141)

**Table 4 polymers-16-02685-t004:** Average values and standard deviation of area and perimeter measurements of 0.3 mm thick lines printed at different angles.

Parameters	Reference Values	Angle of Printed Lines		
		0°	45°	90°
Perimeter [mm]	10.6	16.25 (±0.603)	18.43 (±0.347)	19.81 (±0.305)
Area [mm^2^]	1.5	2.01 (±0.067)	1.96 (±0.015)	2.00 (±0.028)

**Table 5 polymers-16-02685-t005:** Average values and standard deviation of dot roundness.

Sample	Roundness	
	Without Thermal Activation	Thermally Activated
HS	0.954 (±0.012)	0.910 (±0.017)
LS	0.957 (±0.016)	0.951 (±0.019)

## Data Availability

The data are contained in this article.

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
