# Peer review of "Impact of Shape Transformation of Programmable 3D Structures on UV Print Quality"

_polymers, 2024, doi:10.3390/polym16192685_

Round 1
Reviewer 1 Report
Comments and Suggestions for Authors
The paper is well-written, and the results are presented effectively. However, I have some general questions regarding the novelty and potential real-world applications of this research:
-
Can the authors provide specific examples of real-world applications where this type of 4D printing could be effectively used? It would help to better understand the potential impact of this research.
-
Is there any method to control the deformation of the printed structure after it has been programmed? If not, how could we design and utilize such structures for future applications without control over their behavior?
-
How do the authors think certain parameters might affect the mechanical properties of the printed structures? Should mechanical properties be considered when evaluating the performance and usability of these structures?
Author Response
Author's Reply to the Review Report (Reviewer 1)
Can the authors provide specific examples of real-world applications where this type of 4D printing could be effectively used? It would help to better understand the potential impact of this research.
Thank you for your question.
Color-printed 3D programmable structures can be effectively utilized in various practical applications. For example, they can be incorporated into consumer products like customizable phone holders and hangers, where both functionality and aesthetic appeal are key. In interior design, these structures can serve as decorative elements that change shape or color over time, adding dynamic and interactive features to living spaces. Additionally, in education, color-enhanced programmable structures can be used as didactic tools to visually demonstrate concepts through shape transformation, thereby enhancing learning experiences. These applications underscore the versatility and impact of 4D-printed, color-enhanced programmable structures in everyday life.
At the end of the Introduction, we add the following text:
This study opens new possibilities for refining material stress programming and ink printing techniques, thereby expanding the scope of 4D printing applications. Color-printed 3D programmable structures can be utilized in consumer products, such as customizable phone holders and hangers, as well as in interior design, educational tools, and more. These applications combine functionality with aesthetics, providing innovative and adaptive solutions.
Is there any method to control the deformation of the printed structure after it has been programmed? If not, how could we design and utilize such structures for future applications without control over their behavior?
As a method for controlling and predicting the angle of transformation or deformation of printed structures, empirical models can be used. These models can effectively predict the required transformation time to achieve the desired transformation angle, depending on the composition of the active element, its length, and the activation temperature.
How do the authors think certain parameters might affect the mechanical properties of the printed structures? Should mechanical properties be considered when evaluating the performance and usability of these structures?
Thank you for your insightful question. The mechanical properties of printed structures are significantly influenced by various parameters, including material composition, printing temperature, print sequence, and thermal activation. These factors affect not only the tensile strength, elasticity, and rigidity of the materials, but also their behavior under deformation. For instance, in the case of PLA, thermal activation leads to increased stiffness and strength due to molecular rearrangements, while materials like TPU remain flexible and exhibit high tensile strain even after activation.
Mechanical properties must indeed be considered when evaluating the performance and usability of 4D-printed structures. For example, proper material selection and control of interfacial adhesion are crucial for preventing delamination, especially when combining materials with different thermal expansion coefficients, such as PLA and ABS. These properties are critical for ensuring the long-term stability and functionality of 4D-printed products, particularly in applications where both strength and flexibility are required, such as in multi-material programmable components. We agree that mechanical properties are crucial for evaluating the overall functionality and reliability of these structures and have addressed this consideration in the mentioned articles. [1, 2]
- Pivar, M.; Gregor-Svetec, D.; Muck, D. Effect of Printing Process Parameters on the Shape Transformation Capability of 3D Printed Structures. Polymers 2022, 14, 117. https://doi.org/10.3390/polym14010117
- Pivar, M.; Vrabič-Brodnjak, U.; Leskovšek, M.; Gregor-Svetec, D.; Muck, D. Material Compatibility in 4D Printing: Identifying the Optimal Combination for Programmable Multi-Material Structures. Polymers 2024, 16, 2138. https://doi.org/10.3390/polym16152138

Reviewer 2 Report
Comments and Suggestions for Authors
1. abstract should include additional results.
2. Literature review part should be added in the introduction section and they prefer discussed the previous studies critically to show the gap.
3. The novelty statement added to the end of the introduction.
4. References are needed to support the equation of the deformation percentage of the samples dimensions in page 4.
5. References are needed to support the equation to measure the CIELAB colour values in page 5.
6. References are needed to support the equation to determine the dot roundness in page 6.
7. Page 7 , raw 7 references are needed to support this statement (The smaller dimensional changes ……. dimensional changes in the 3D printed samples during thermal activation.).
8. Page 7 , raw 14 references are needed to support these statements (The extent of the shrinkage of the 3D samples is …….. The deviations in the shrinkage of the 3D samples are due to the thickness of the UV ink films.
9. The discussion of the results must be based on facts (such as optical inspections) or supported by references; therefore, these statements should be modified (A stronger expansion after shrinking can be observed with CMYK …….. in the elongation of the 3D printed samples.)
10. All results and discussion that mentioned in Section 3 should be supported by references.
11. Section 4 included details of the (Conclusion) only; therefore the title of this section should be changed from (Discussion) to (Conclusion). And the title of Section 3 should be changed from (Results) to (Results and Discussion).

Moderate editing of English language required.
Author Response
Author's Reply to the Review Report (Reviewer 2)
Abstract should include additional results.
Thank you. We have completely rewritten the abstract and included additional results.
The field of 3D and 4D printing is advancing rapidly, offering new ways to control the transformation of programmable 3D structures in response to external stimuli. This study examines the impact of 3D printing parameters, UV ink thickness (applied using a UV inkjet printer on pre-3D-printed programmable structures), and thermal activation on the dimensional and surface changes of high-stress (HS) and low-stress (LS) programmable samples and on print quality. Results indicate that HS samples shrink in the longitudinal direction while expanding in height and width, whereas LS samples exhibit minimal dimensional changes due to lower programmed stress. Dynamic mechanical analysis shows that UV ink, particularly cyan and CMYK overprints, reduces shrinkage in HS samples by acting as a resistive layer. Thicker ink films further reduce dimensional changes in HS samples. Thermal activation increases surface roughness in HS structures, leading to wrinkling of UV ink films, while LS structures are less affected. Surface gloss decreases significantly in HS structures after UV ink application, but thermal activation has little impact on LS structures. UV ink adhesion remains strong across both HS and LS samples, suggesting that UV inks are ideal for printing on programmable 3D structures where color print quality and precise control of shape transformation are crucial.
Literature review part should be added in the introduction section and they prefer discussed the previous studies critically to show the gap.
We have revised the Introduction to include a more detailed literature review discussing key studies related to surface roughness, gloss, and ink adhesion in both 3D and 4D printing. We critically addressed the impact of these factors on print quality, with references to relevant studies, including the challenges of maintaining ink adhesion during the shape transformation process in 4D printing. We also identified a gap in understanding the interaction between residual stress and ink adhesion post-thermal activation, which our study aims to address. We hope this revision meets the expectations for a critical review of prior research and highlights the research gap effectively.
[23] Sang, R., Manley, A. J., Wu, Z., Feng, X. Digital 3D Wood Texture: UV-Curable Inkjet Printing on Board Surface. Coatings 2020,10, 1144. doi:10.3390/coatings10121144
[26] Scotton, R., Guerrini, L. M., Oliveira, M. P. Evaluation of solvent-based and UV-curing inkjet inks on the adhesion and printing quality of different aircraft surfaces coating. Progress in Organic Coatings 2021, 158, 106389. https://doi.org/10.1016/j.porgcoat.2021.106389.
[29] Sang, R.J.; Yang, S.Q.; Fan, Z.X. Effects of MDF Substrate Surface Coating Process on UV Inkjet Print
Quality. Coatings 2023, 13, 970. https://doi.org/10.3390/coatings13050970.
[30] Plazonic, I.; Bates, I.; Barbaric-Mikocevic, Z. The Effect of Straw Fibers in Printing Papers on Dot Reproduction Attributes, as Realized by UV Inkjet Technology. BioResources 2016, 11, https://doi.org/10.15376/biores.11.2.5033-5049
The novelty statement added to the end of the introduction.
At the end of the Introduction, we add the following text:
This study opens new possibilities for refining material stress programming and ink printing techniques, thereby expanding the scope of 4D printing applications. Color-printed 3D programmable structures can be utilized in consumer products, such as customizable phone holders and hangers, as well as in interior design, educational tools, and more. These applications combine functionality with aesthetics, providing innovative and adaptive solutions.
References are needed to support the equation of the deformation percentage of the samples dimensions in page 4.
Thank you for your suggestion. We have added two references to support the equation of the deformation percentage of the sample dimensions on page 4.
[39] Wang, F.; Luo, F.; Huang, Y.; Cao, X.; Yuan, C. 4D printing via multispeed fused deposition modeling. Advanced Materials Technologies 2023, 8(2), 2201383. https://doi.org/10.1002/admt.202201383
[40] Ansaripour, A.; Heidari-Rarani, M.; Mahshid, R.; Bodaghi, M. Influence of extrusion 4D printing parameters on the thermal shape-morphing behaviors of polylactic acid (PLA). The International Journal of Advanced Manufacturing Technology 2024, 132(3), 1827-1842. https://doi.org/10.1007/s00170-024-13470-6
References are needed to support the equation to measure the CIELAB colour values.
Thank you for your feedback. We have added five relevant references to support the equation used to measure the CIELAB color values, as requested.
[26] Scotton, R. S.; Guerrini, L. M.; Oliveira, M. P. Evaluation of solvent-based and UV-curing inkjet inks on the adhesion and printing quality of different aircraft surfaces coating. Progress in Organic Coatings 2021, 158, 106389. https://doi.org/10.1016/j.porgcoat.2021.106389.
[41] Wojciech, M.; Maciej, T. Color Difference Delta E - A Survey. Machine Graphics and Vision 2011, 20.
References are needed to support the equation to determine the dot roundness in page 6.
Thank you for your feedback. We have added two references to support the equation to determina the dot roundness in page 6, as requested.
[30] Plazonic, I.; Bates, I.; Barbaric-Mikocevic, Z. The Effect of Straw Fibers in Printing Papers on Dot Reproduction Attributes, as Realized by UV Inkjet Technology. BioResources 2016, 11, https://doi.org/10.15376/biores.11.2.5033-5049
[45] T. Ferreira, W.R. ImageJ User Guide. IJ 1.46r National Institutes of Health, Washington. 2012.
Page 7 , raw 7 references are needed to support this statement (The smaller dimensional changes ……. dimensional changes in the 3D printed samples during thermal activation).
Thank you. The text has been revised and rewritten with newly added references.
The smaller dimensional changes of the LS samples are influenced by several factors. One factor is the isotropic structure of the 3D-printed samples, which is created by a rectilinear infill pattern. Wang et al. [46] reported that a rectilinear infill pattern eliminates the differences in shrinkage rate and thus minimises the transformation. Other factors that reduce the dimensional changes of 3D printed samples are the 3D printing parameters. The higher the 3D printing temperature and build plate temperature and the lower the printing speed, the less prestress is stored in the material and the lower the dimensional changes of the 3D printed sample [39,47–49].
[46] Wang, G.; Tao, Y.; Capunaman, O. B., Yang, H.; Yao, L. A-line: 4D printing morphing linear composite structures. In Proceedings of the 2019 CHI Conference on Human Factors in Computing Systems 2019 (pp. 1-12). https://doi.org/10.1145/3290605.3300656
[39] Wang, F.; Luo, F.; Huang, Y.; Cao, X.; Yuan, C. 4D printing via multispeed fused deposition modeling. Advanced Materials Technologies 2023, 8(2), 2201383. https://doi.org/10.1002/admt.202201383
[47] Wang, Y.; Li, X. 4D-Printed Bi-Material Composite Laminate for Manufacturing Reversible Shape-Change Structures. Compos B Eng 2021, 219, 108918. https://doi.org/10.1016/j.compositesb.2021.108918
[48] Van Manen, T.; Janbaz, S.; Zadpoor, A.A. Programming 2D/3D Shape-Shifting with Hobbyist 3D Printers. Mater Horiz 2017, 4, 1064-1069. https://doi:10.1039/c7mh00269f
[49] Hu, G.; Bodaghi, M. Direct Fused Deposition Modeling 4D Printing and Programming of Thermoresponsive Shape Memory Polymers with Autonomous 2D-to-3D Shape Transformations. Adv Eng Mater 2023, 25, 2300334. https://doi:10.1002/adem.202300334.
Page 7 , raw 14 references are needed to support these statements (The extent of the shrinkage of the 3D samples is …….. The deviations in the shrinkage of the 3D samples are due to the thickness of the UV ink films.
Thank you, the statement has been deleted, and the text has been rewritten.
The discussion of the results must be based on facts (such as optical inspections) or supported by references; therefore, these statements should be modified (A stronger expansion after shrinking can be observed with CMYK …….. in the elongation of the 3D printed samples.)
Thank you, the statement has been deleted, and the text has been rewritten.
All results and discussion that mentioned in Section 3 should be supported by references.
Thank you. The text has been revised, and the reference has been added on pages 13 and 14.
3.5. Colour Differences
The colour differences (∆E*ab) shown in Figure 9 represent the difference between the printed colours and the colour of unprinted material surface before and after thermal activation. In all cases, the average colour difference for the LS samples were between 1.4 and 1.6. This means that the activation medium and temperature do not have a major influence on the appearance of the colours. These colour differences are very small and normally indistinguishable to an inexperienced observer, but they are visible to the trained eye when the samples are placed side by side [26,41]. A slightly higher deviation of the average values between the colours was observed in the HS samples. The values of the colour differences range from 0.6 to 2.6. The largest colour differences of 2.6 were observed for the colour cyan. These colour differences are obvious to the untrained human eye (2< ∆E*ab <3.5) [41], which means that shrinking the HS samples changes the appearance of the colour. Other colour differences are small and cannot be distinguished by an untrained eye.
[41] Wojciech, M.; Maciej, T. Color Difference Delta E - A Survey. Machine Graphics and Vision 2011, 20, 383-411.
UV inkjet printing is characterised by the fact that the UV ink changes almost instantly from a liquid to a solid form when exposed to sufficient UV radiation [52].
[52] Simon, J.; Langenscheidt, A. Curing Behavior of a UV-Curable Inkjet Ink: Distinction between Surface-Cure and Deep-Cure Performance. J Appl Polym Sci 2020, 137, 49218. https://doi:10.1002/app.49218.
This indicates that a physical gain has occurred due to the spreading of the liquid UV ink before curing due to the low surface tension of the material [53] and the scattering of a very small satellite ink droplet at the edge of the fine element [30].
[53] Rotar, B.; Elesini, U.S.; Hajdu, P.; Leskovar, B.; Urbas, R. Morphological and Dimensional Properties of Unmodified and Modified Braille Dots Produced with UV Inkjet Printing. Materiali in Tehnologije 2020, 54, https://doi:10.17222/mit.2020.016
[30] Plazonic, I.; Bates, I.; Barbaric-Mikocevic, Z. The Effect of Straw Fibers in Printing Papers on Dot Reproduction Attributes, as Realized by UV Inkjet Technology. BioResources 2016, 11, https://doi.org/10.15376/biores.11.2.5033-5049
Section 4 included details of the (Conclusion) only; therefore the title of this section should be changed from (Discussion) to (Conclusion). And the title of Section 3 should be changed from (Results) to (Results and Discussion).
Thank you for your valuable suggestions. We have changed the titles of the sections as you recommended.
